

# NADPH-dependent thioredoxin reductase C plays a role in nonhost disease resistance against *Pseudomonas syringae* pathogens by regulating chloroplast-generated reactive oxygen species

Yasuhiro Ishiga[1,2], Takako Ishiga[2], Yoko Ikeda[3], Takakazu Matsuura[3] and Kirankumar S. Mysore[2]

[1] Faculty of Life and Environmental Sciences, University of Tsukuba, Tsukuba, Japan
[2] Plant Biology, The Samuel Roberts Noble Foundation, Ardmore, OK, USA
[3] Institute of Plant Science and Resources, Okayama University, Kurashiki, Japan

Corresponding author
Yasuhiro Ishiga,
ishiga.yasuhiro.km@u.tsukuba.ac.jp

## ABSTRACT

Chloroplasts are cytoplasmic organelles for photosynthesis in eukaryotic cells. In addition, recent studies have shown that chloroplasts have a critical role in plant innate immunity against invading pathogens. Hydrogen peroxide is a toxic by-product from photosynthesis, which also functions as a signaling compound in plant innate immunity. Therefore, it is important to regulate the level of hydrogen peroxide in response to pathogens. Chloroplasts maintain components of the redox detoxification system including enzymes such as 2-Cys peroxiredoxins (2-Cys Prxs), and NADPH-dependent thioredoxin reductase C (NTRC). However, the significance of 2-Cys Prxs and NTRC in the molecular basis of nonhost disease resistance is largely unknown. We evaluated the roles of Prxs and NTRC using knock-out mutants of *Arabidopsis* in response to nonhost *Pseudomonas syringae* pathogens. Plants lacking functional NTRC showed localized cell death (LCD) accompanied by the elevated accumulation of hydrogen peroxide in response to nonhost pathogens. Interestingly, the *Arabidopsis ntrc* mutant showed enhanced bacterial growth and disease susceptibility of nonhost pathogens. Furthermore, the expression profiles of the salicylic acid (SA) and jasmonic acid (JA)-mediated signaling pathways and phytohormone analyses including SA and JA revealed that the *Arabidopsis ntrc* mutant shows elevated JA-mediated signaling pathways in response to nonhost pathogen. These results suggest the critical role of NTRC in plant innate immunity against nonhost *P. syringae* pathogens.

# INTRODUCTION

Plants are surrounded by a large number of microbes, including potential pathogens, in their natural habitat. Thus, plants have evolved specific immune systems to defend

themselves against invading pathogens by developing a wide variety of constitutive and inducible defenses. Constitutive defenses include preformed defense structures including cell walls, surface waxes, the position of stomata, and preformed antimicrobial compounds. In addition to constitutive defenses, plants have developed multiple layers of advanced surveillance systems against invading pathogens to activate a wide array of inducible defense responses including rapid oxidative burst, callose ($\beta$-1,3-glucan) deposition in the cell wall, the induction of hormone-mediated signaling pathways leading to the expression of defense-related genes, and the production of antimicrobial compounds (*Hok, Attard & Keller, 2010*). To date, these surveillance systems are known to include two layers of plant immune responses against invading pathogens. The first layer of plant immune responses is pathogen-associated molecular pattern (PAMP)-triggered immunity (PTI), which recognizes the conserved molecules from invading pathogens (microbes) using plant pattern-recognition receptors (PRRs) (*Zipfel, 2008*). The second layer of plant immune responses that exists to defend themselves from invading pathogens is effector-triggered immunity (ETI) based on the gene-for-gene interactions, which are governed by individual plant resistance (*R*) genes and their corresponding avirulence (*Avr*) genes (*Cui, Tsuda & Parker, 2014*). ETI is often associated with the hypersensitive response (HR). The HR involves a wide array of responses including the activation of defense-associated genes, such as pathogenesis-related (*PR*) genes, phytoalexin production, and the formation of rapid localized cell death (LCD) at the site of pathogen invasion (*Coll, Epple & Dangl, 2011*). The HR-mediated LCD is the well-characterized form of plant programmed cell death (PCD) (*Greenberg & Yao, 2004*). Although the precise role of death signals associated with the LCD is still unknown, the most investigated signal for LCD is reactive oxygen species (ROS) (*Mur et al., 2008*). Apoplastic ROS derived from plasma membrane-bound NADPH oxidase is well known to regulate LCD (*Mur et al., 2008*). In addition to apoplastic ROS, chloroplast-derived ROS have been shown to function in LCD (*Ishiga et al., 2009*; *Zurbriggen et al., 2009*; *Ishiga et al., 2012*). *Zurbriggen et al. (2009)* reported the requirement of chloroplast-generated ROS for LCD, based on tobacco plants over-expressing a chloroplast-targeted cyanobacterial flavodoxin. *Lim et al. (2010)* demonstrated that loss-of-function analysis of *SlFTR-c* (*ferredoxin:thioredoxin reductase-c*) resulted in spontaneous necrosis development associated with elevated accumulation of ROS without pathogen inoculation.

The chloroplast is the organelle which conducts photosynthesis to fix carbon in nature. In addition, chloroplasts carry out a number of other functions, including fatty acid biosynthesis and amino acid biosynthesis (*Rolland et al., 2012*). The biosynthesis of plant immunity-related phytohormones including SA and JA occurs in the chloroplast (*Wasternack & Hause, 2013*; *Seyfferth & Tsuda, 2014*). It was demonstrated that PTI signals are transmitted to the chloroplast and activate $Ca^{(2+)}$ signaling in the stroma through a calcium-sensing receptor (*Nomura et al., 2012*). A recent study demonstrated that chloroplasts extend chloroplast stromules to nuclei during plant immunity as well as reactive oxygen species (ROS) stress (*Brunkard, Runkel & Zambryski, 2015*; *Caplan et al., 2015*). Therefore, chloroplasts have a critical role in plant immunity against pathogens. Furthermore, the chloroplast produces retrograde signals, such as ROS, to regulate the nuclear gene expression in order to modulate chloroplast

biogenesis, maintain homeostasis, or optimize chloroplast performance under stress conditions, including pathogen attacks (*Chi, Sun & Zhang, 2013*; *Chi et al., 2015*). Low levels of ROS are known to function as a retrograde signal, whereas excessive levels of ROS are known to cause oxidative damage resulting in LCD (*Hossain et al., 2015*). Therefore, plants have developed suitable mechanisms to modulate the excessive production of ROS in the chloroplast. A chloroplast ROS detoxification mechanism includes several antioxidant enzymes such as ascorbate peroxidase (APX), glutathione peroxidase (GPX), and peroxiredoxin (Prx) (*Gill & Tuteja, 2010*). Prxs are known to function together with NADPH dependent thioredoxin reductase C(NTRC), which is a central player in the chloroplast redox detoxification system in *Arabidopsis* plants (*Kirchsteiger et al., 2009*; *Bernal-Bayard et al., 2012*; *Bernal-Bayard et al., 2014*; *Puerto-Galán et al., 2015*). It has been shown that there are four chloroplast Prx isoforms in *Arabidopsis*, including PrxA, PrxB, PrxQ, and PrxIIE (*Tripathi, Bhatt & Dietz, 2009*).

We have previously shown that loss-of-function analysis of *Prx* and *NTRC* resulted in accelerated *Pseudomonas syringae* pv. tomato DC3000 (*Pto* DC3000) disease-associated cell death with enhanced ROS accumulation in tomatoes and *Arabidopsis* (*Ishiga et al., 2012*). Our previous studies also demonstrated that *Pto* DC3000 targets the chloroplast ROS homeostasis and enhances ROS accumulation by suppressing the expression of genes encoding chloroplast ROS detoxification enzymes including APX, Prx, and NTRC in *Arabidopsis* and tomato during pathogenesis (*Ishiga et al., 2009*; *Ishiga et al., 2012*). This indicates that the NTRC/Prx chloroplast ROS detoxification system functions as a negative regulator of disease-associated cell death.

Nonhost resistance is known as the most common and durable form of plant disease resistance against almost all microbes (including parasites and pathogens) in nature. Nonhost resistance is believed to be regulated by many genes and to be involved in multiple layers, including constitutive and inducible defense responses such as PTI and ETI (*Mysore & Ryu, 2004*; *Niks & Marcel, 2009*; *Fan & Doerner, 2012*; *Senthil-Kumar & Mysore, 2013*). However, nonhost resistance is one of the least understood forms of resistance and is believed to be more durable than host specific resistance in the field (*Gill, Lee & Mysore, 2015*). The molecular basis of host specific resistance has been well characterized. Unlike host resistance, a limited number of players underlying nonhost resistance have been identified (*Fan & Doerner, 2012*; *Senthil-Kumar & Mysore, 2013*). Therefore, an understanding of the molecular basis of nonhost resistance may provide an avenue to engineer crop plants to confer durable resistance against a wide range of pathogens.

Our previous results have shown that chloroplast ROS production plays a role in regulating disease-associated cell death in tomato and *Arabidopsis* (*Ishiga et al., 2009*; *Ishiga et al., 2012*). However, the precise role of chloroplast-derived ROS associated with LCD is still unclear. To further understand the role of chloroplast-derived ROS in the interactions of plants and pathogens, we investigated the functional analysis of the NTRC/Prx chloroplast ROS detoxification system during nonhost resistance in *Arabidopsis* against *Pseudomonas syringae*, and demonstrated that the *ntrc* mutant is compromised in nonhost resistance. Expression profiles of SA- and JA-mediated signaling pathways and phytohormone analyses including SA and JA identified a correlation between the JA-mediated signaling pathway

and the enhanced susceptibility of the *ntrc* mutant against nonhost *P. syringae*. Thus, our results suggest the importance of the NTRC/Prx chloroplast ROS detoxification system in plant immunity.

## MATERIALS AND METHODS

### Plant materials and bacterial strains

*Arabidopsis thaliana* ecotype Columbia (Col-0) was used as a wild-type plant in this study. *A. thaliana* homozygous T-DNA insertion mutants *prxA* (At3g11630; CS875813), *prxB* (At5g06290; SALK_017213), *prxQ* (At3g26060; SALK_070860), and *prxIIE* (At3g52960; SALK_064512) were identified from the SALK Institute's collection (*Ishiga et al., 2012*). Seeds of the *Arabidopsis ntrc* T-DNA mutant and its complemented line (*Perez-Ruiz et al., 2006*) were kindly provided by Dr. Francisco Javier Cejudo (Instituto de Bioquímica Vegetal y Fotosíntesis, Spain). Seeds of *Arabidopsis* plants were germinated and maintained on 1/2 Murashige and Skoog (MS) medium (0.3% phytagel) with Gamborg vitamins (Sigma-Aldrich, St. Louis, MO, USA) and used for inoculation assays two weeks after germination at 25 °C with a light intensity of 200 $\mu$E m$^{-2}$s$^{-1}$ and a 12 h light/12 h dark photoperiod.

Nonhost pathogens *Pseudomonas syringae* pv. *tabaci* (*Pta*) 6605 (*Taguchi et al., 2001*), *P. syringae* pv. *glycinea* (*Pgl*) race 4 (*Staskawicz, Dahlbeck & Keen, 1984*), and *P. syringae* pv. *tomato* (*Pto*) T1 (*Almeida et al., 2009*) were used to study nonhost resistance. *P. syringae* strains were grown at 28 °C on mannitol-glutamate (MG) medium (*Keane, Kerr & New, 1970*) for 36–48 h. Prior to inoculation, bacteria were suspended in sterile distilled $H_2O$ and bacterial cell densities ($OD_{600}$) were measured using a JASCO V-730 spectrophotometer (JASCO, Tokyo, Japan).

### Seedling flood-inoculation method

*Arabidopsis* seedlings were inoculated by a method where MS agar plates were flooded with bacterial cells as described previously (*Ishiga et al., 2011*). To perform uniform inoculation, 40 ml of bacterial suspension made in sterile distilled $H_2O$ containing 0.025% Silwet L-77 (OSI Specialties Inc., Danbury, CT, USA.) was dispensed into the plate containing 2-week-old *Arabidopsis* seedlings, and the plates were incubated for 2–3 min at room temperature. After the bacterial suspension was removed by decantation, the plates containing inoculated plants were sealed with 3M Micropore 2.5 cm surgical tape (3M, St. Paul, MN, USA) and incubated at 24 °C with a light intensity of 150–200 $\mu$E m$^{-2}$ s$^{-1}$ and a 12 h light/12 h dark photoperiod. Symptom development was observed at 2 dpi. In each experiment, 8 plants were evaluated, and each experiment was repeated at least three times.

To determine the bacterial growth in *Arabidopsis* leaves, we measured the internal bacterial population at 2 dpi. For the determination of internal bacterial growth, inoculated seedlings were collected by cutting the hypocotyls to separate the above agar parts (whole rosette) from the Phytagel plate, and the total weight of the inoculated seedlings was measured. After measurement of the seedlings' weight, the seedlings were surface-sterilized with 5% $H_2O_2$ for 3 min. After washing three times with sterile distilled water, the plants were homogenized in sterile distilled water, and the diluted samples were plated onto MG
medium. Two days after plating of the diluted samples, the bacterial colony forming units (CFU) were counted using proper diluted samples. The CFU was normalized as CFU/mg using total weights of the inoculated seedlings. Bacterial populations were evaluated in three independent experiments.

### Detection of cell death

Cell death was estimated by measuring ion leakage from five plants treated with water (mock) or inoculated with nonhost *P. syringae* pathogens and incubated at 24 °C with a light intensity of 200 $\mu$E m$^{-2}$ s$^{-1}$ and a 12 h light/12 h dark photoperiod as described previously (*Ishiga et al., 2012*). The inoculated plants were collected and then gently agitated in 30 ml of distilled water for 3 h, and the leachates were measured using an ion conductivity meter (Omron, Kyoto, Japan). The plants were then autoclaved for 20 min to kill the cells and release the total ions into the water. The values relative to the whole ion content after autoclaving were used to express the percent ion leakage.

### Detection of hydrogen peroxide

The generation of hydrogen peroxide was detected using 3,3′-diaminobenzidine (DAB) staining as described previously (*Ishiga et al., 2009*; *Ishiga et al., 2012*; *Rojas et al., 2012*). To quantify the accumulation of hydrogen peroxide over time, 10 leaves were collected at 24 h and 48 h after inoculation with nonhost *P. syringae* pathogens, and then the leaves were placed in 1 mg/ml DAB-HCl (pH 3.8). After incubation for 6 h at room temperature, chlorophyll was removed with 95% ethanol and the leaves were mounted in 50% glycerol. DAB staining in the leaves was quantified using Image J software (version 1.49). The intensity of staining was expressed as a percentage of coloration, where the intensity of coloration of the mock-inoculated wild-type Col-0 was set to 100%.

### Quantification of phytohormones

Approximately 100 mg of fresh weight of two-week-old *Arabidopsis* plants grown on MS plates were used for extraction. Extraction, purification, and quantification were performed as described by *Tsukahara et al. (2015)*. Quantification was performed by an Agilent 1260-6410 Triple Quad LC/MS (Agilent Technologies Inc., Santa Clara, CA, USA) equipped with a ZORBAX Eclipse XDB-C18 column (Agilent Technologies Inc.) using four independent samples for each genotype.

### Real-time quantitative RT-PCR

Total RNA extraction and real-time quantitative RT-PCR (qRT-PCR) were done as described previously (*Ishiga et al., 2013*). Total RNA was extracted using RNAiso Plus (TaKaRa, Otsu, Shiga, Japan) according to the manufacturer's protocol. Two $\mu$g of total RNA was treated with gDNA Eraser (TaKaRa) to eliminate genomic DNA, and the DNase-treated RNA was reverse transcribed using the PrimeScript$^{TM}$ RT reagent Kit (TaKaRa). The cDNA (1:20) was then used for qRT-PCR which was performed using the primers shown in Table S1 with SYBR$^{®}$ Premix Ex Taq$^{TM}$ II (TaKaRa) on a Thermal Cycler Dice$^{®}$ Real Time System (TaKaRa). The *Arabidopsis UBIQUITIN EXTENSION PROTEIN 1* (*UBQ1*) was used as an internal control to normalize gene expression. The average CT

values calculated using the 2nd derivative maximum method from triplicate samples were used to determine the fold expression relative to the controls.

## RESULTS

### An *Arabidopsis ntrc* mutant shows accelerated cell death in response to nonhost *Pseudomonas syringae* pathovars

To investigate the role of chloroplast ROS homeostasis during nonhost resistance, we analyzed four individual T-DNA insertional mutants deficient in chloroplast-localized peroxiredoxins (*PrxA*, *PrxB*, *PrxQ*, and *PrxIIE*) and a T-DNA insertion mutant of *NTRC*, an electron donor for Prxs in the Trx system (*Tripathi, Bhatt & Dietz, 2009*; *Cejudo et al., 2012*). Two-week-old *Arabidopsis* plants including Col-0 (wild-type), and the *prx* and *ntrc* mutants grown on MS agar were flood-inoculated (*Ishiga et al., 2011*) with the non-host pathogens *Pta* 6605 (*Taguchi et al., 2001*), *Pgl* race 4 (*Staskawicz, Dahlbeck & Keen, 1984*), and *Pto* T1 (*Almeida et al., 2009*). Nonhost bacterial pathogens in *Nicotiana benthamiana* induce two categories of nonhost resistance responses: Type I, which does not result in visible cell death; and Type II, which includes LCD in response to nonhost pathogens (*Mysore & Ryu, 2004*; *Oh et al., 2006*). We therefore first monitored for the Type I and Type II response to nonhost pathogens in wild-type plants. *Pta* 6605 induced cell death which was associated with a high percentage of ion leakage in the wild-type Col-0 (Figs. 1A and 1B). On the other hand, wild-type Col-0 plants inoculated with nonhost pathogens *Pgl* race 4 and *Pto* T1 showed no obvious symptoms or ion leakage, and remained healthy (Figs. 1A and 1B). These results indicate that *Pta* 6605 involves the Type I nonhost resistance, whereas *Pgl* race 4 and *Pto* T1 involve the Type II nonhost resistance in *Arabidopsis*. No significant difference was observed in symptom development and ion leakage between the wild-type Col-0 and the *prxa*, *prxb*, *prxq*, and *prxIIe* mutants in response to *Pta* 6605, *Pgl* race 4, and *Pto* T1 (Figs. 1A and 1B).

Interestingly, the knock-out mutant of *ntrc* showed accelerated cell death in response to *Pta* 6605 when compared to the wild-type Col-0 (Fig. 1A). Surprisingly, the type II nonhost pathogens *Pgl* race 4 and *Pto* T1 also induced cell death on *ntrc* plants, unlike the wild-type Col-0 and the *prx* mutants (Fig. 1A). Consistent with the visible phenotypes, a higher percentage of ion leakage was observed in *ntrc* mutant plants inoculated with non-host pathogens; however, *Pta* 6605 induced slightly stronger cell death in the *ntrc* mutant than the other nonhost pathogens tested (Fig. 1B).

### An *Arabidopsis ntrc* mutant shows enhanced susceptibility to nonhost *Pseudomonas syringae* pathovars

To test if the severe disease symptoms are associated with increased bacterial multiplication in the *ntrc* mutant, the bacterial populations were monitored two days after inoculation with the nonhost pathogens *Pta* 6605, *Pgl* race 4, and *Pto* T1. The bacterial populations of *Pta* 6605, *Pgl* race 4, and *Pto* T1 in the *ntrc* mutant were higher than those in the wild-type Col-0 and the *prx* mutants (Figs. 1C–1E). Interestingly, the *ntrc* mutant inoculated with *Pgl* race 4 showed significantly increased bacterial populations compared with the other nonhost pathogens (Figs. 1C–1E). Furthermore, the complement line for the *ntrc* mutant

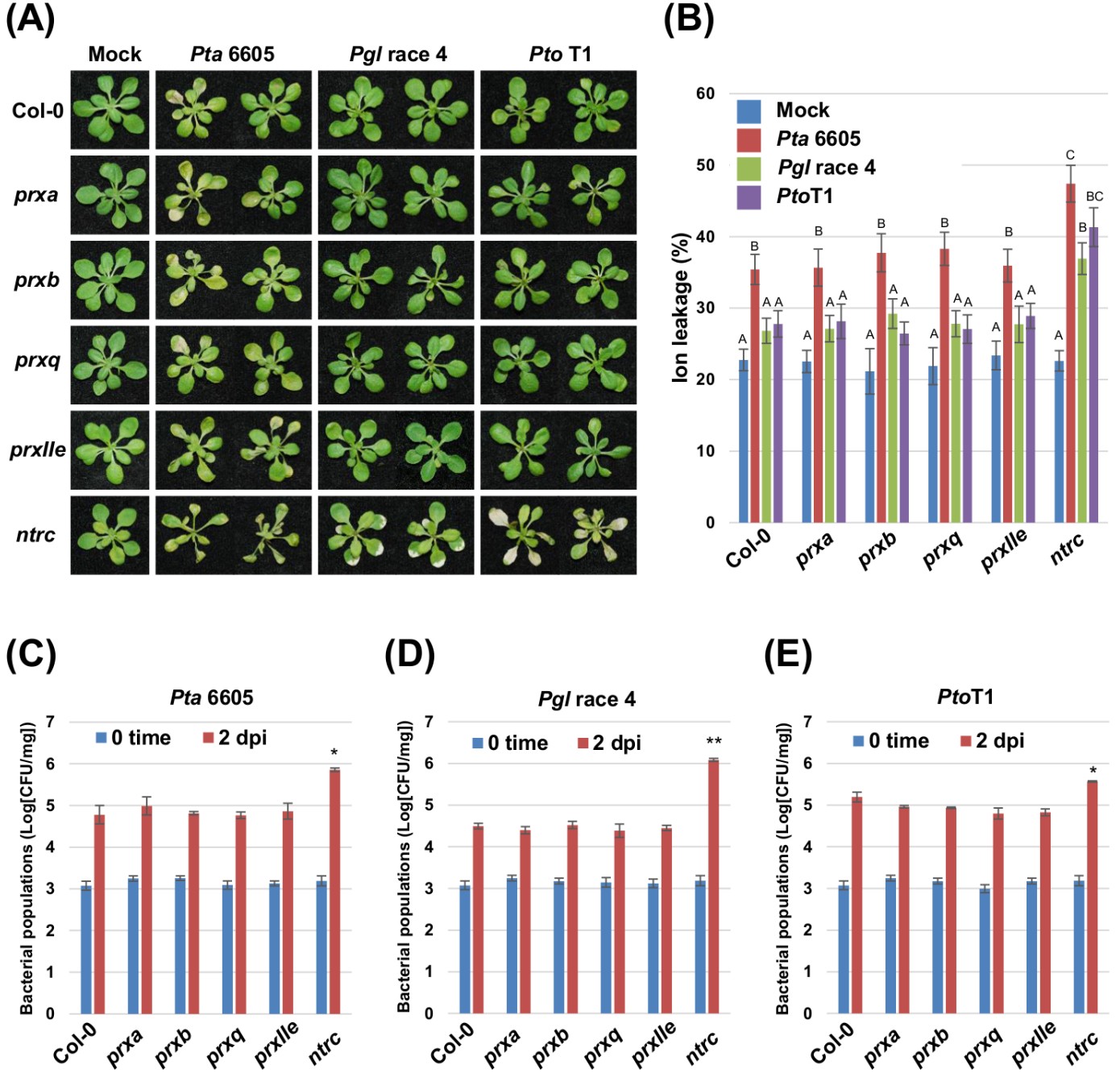

**Figure 1** **Analysis of the nonhost resistance responses of the *Arabidopsis thaliana* wild-type Col-0, and the *prx* (*prxa*, *prxb*, *prxq* and *pexIIe*) and *ntrc* mutants using a seedling flood-inoculation assay.** (A) Response of the *Arabidopsis thaliana* wild-type Col-0, and the *prx* (*prxa*, *prxb*, *prxq*, and *pexIIe*) and *ntrc* mutants to nonhost *Pseudomonas syringae* pathogens. *Arabidopsis* seedlings were flood-inoculated with nonhost pathogens including *Pseudomonas syringae* pv. *tabaci* (*P. s.* pv. *tabaci*), *P. syringae* pv. *glycinea* (*P. s.* pv. *glycinea*), and *P. syringae* pv. *tomato* T1 (*P. s.* pv. *tomato* T1) at a concentration of $5 \times 10^6$ CFU/ml. Photographs were taken at 2 dpi. (B) Ion leakage from the *Arabidopsis thaliana* wild-type Col-0, and the *prx* (*prxa*, *prxb*, *prxq*, and *pexIIe*) and *ntrc* mutants flooded with water (mock) or *Pseudomonas syringae* pv. *tabaci* (*P. s.* pv. *tabaci*), *P. syringae* pv. *glycinea* (*P. s.* pv. *glycinea*), and *P. syringae* pv. *tomato* T1 (*P. s.* pv. *tomato* T1) at a concentration of $5 \times 10^6$ CFU/ml. Samples were collected at 2 dpi. Bars show the percentage of total ions. Vertical bars indicate the standard error for three biological replicates. Statistically significant differences are noted as different alphabet characters based on ANOVA ($p < 0.05$). Bacterial populations of (C) *Pseudomonas syringae* pv. *tabaci* (*P. s.* pv. *tabaci*), (D) *P. syringae* pv. *glycinea* (*P. s.* pv. *glycinea*), and (E) *P. syringae* pv. *tomato* T1 (*P. s.* pv. *tomato* T1) in the *Arabidopsis* wild-type Col-0, and the *prx* (*prxa*, *prxb*, *prxq*, and *pexIIe*) and *ntrc* mutants. Bacterial populations were quantified at 0 and 2 dpi. Vertical bars indicate the standard errors for three independent experiments. Asterisks indicate a significant difference from the wild-type Col-0 in a *t*-test (* = $p < 0.05$, ** = $p < 0.01$).

showed almost identical responses with the wild-type Col-0 against nonhost *P. syringae* pathogens (Figs. S1A–S1C). These results suggest that NTRC plays a role in the nonhost resistance response in *Arabidopsis.*

### An *Arabidopsis ntrc* mutant shows accelerated ROS accumulation in response to nonhost *Pseudomonas syringae* pathovars

To investigate if nonhost pathogen-induced cell death is associated with accelerated ROS accumulation, DAB staining was carried out in the wild-type Col-0, and the *prx* and ntrc mutants after flood-inoculation with *Pta* 6605, *Pgl* race 4, and *Pto* T1. The accumulation of hydrogen peroxide represented by the deposition of brown color was detected in the wild-type Col-0, and the *prx* and *ntrc* mutants in response to nonhost pathogens. However, slightly elevated ROS production was detected in the *ntrc* mutants at 24 h after inoculation with *Pta* 6605 when compared to *Pgl* race 4 and *Pto* T1 (Fig. S2). As expected, a higher level of ROS accumulation was observed in the *ntrc* mutant compared with those in the wild-type Col-0 and the *prx* mutants at 48 h after inoculation with nonhost pathogens (Figs. 2A and 2B). Interestingly, higher ROS production (stronger DAB staining) was detected in the *ntrc* mutants at 48 h after inoculation with *Pta* 6605 and *Pgl* race 4 than with *Pto* T1 (Figs. 2A and 2B). Furthermore, the complemented line for the *ntrc* mutant showed the same level of ROS accumulation as the wild-type Col-0 in response to nonhost *P. syringae* pathogens (Figs. S1D and S1E). These results suggest that the severe cell death induced by nonhost pathogens is associated with the accelerated ROS accumulation in the *ntrc* mutant.

We next investigated the expression profiles of marker genes for chloroplast ROS accumulation, including *STROMAL ASCORBATE PEROXIDASE* (*sAPX*), *THYLAKOIDAL ASCORBATE PEROXIDASE* (*tAPX*), *GLUTATHIONE PEROXIDASE 1* (*GPX1*), and *GLUTATHIONE PEROXIDASE 7* (*GPX7*) in response to inoculation with *Pta* 6605. Two-week-old *Arabidopsis* plants, including the wild-type Col-0, the *ntrc* mutant, and the complemented line (35S-*NTRC*/*ntrc*) were grown on MS plates, and were inoculated with *Pta* 6605, and then total RNA was isolated from samples fixed at 3, 6, 12, and 24 h after inoculation. There was no significant difference among the wild-type Col-0, the *ntrc* mutant, and the complemented line with respect to the expression profiles of *sAPX*, *tAPX*, and *GPX1* (Figs. S3A–S3C). The expression of *GPX7* was only induced in the *ntrc* mutant when compared with the wild-type Col-0 and complemented line (Fig. S3D).

### An *Arabidopsis ntrc* mutant shows elevated JA-mediated signaling pathways in response to *Pta* 6605

The phytohormone-mediated signaling pathways leading to defense responses against invading pathogens have been shown to have a critical role in plant immunity (*Robert-Seilaniantz, Grant & Jones, 2011*; *Pieterse et al., 2012*). In general, the SA-mediated signaling pathway is implicated in the regulation of defense responses against biotrophic and hemibiotrophic pathogens, while the JA pathway is associated with defense responses against necrotrophic pathogens (*Robert-Seilaniantz, Grant & Jones, 2011*; *Pieterse et al., 2012*). Since several studies have demonstrated that phytohormone-mediated signaling pathways play a role in nonhost resistance against fungal and bacterial pathogens

**(A)**

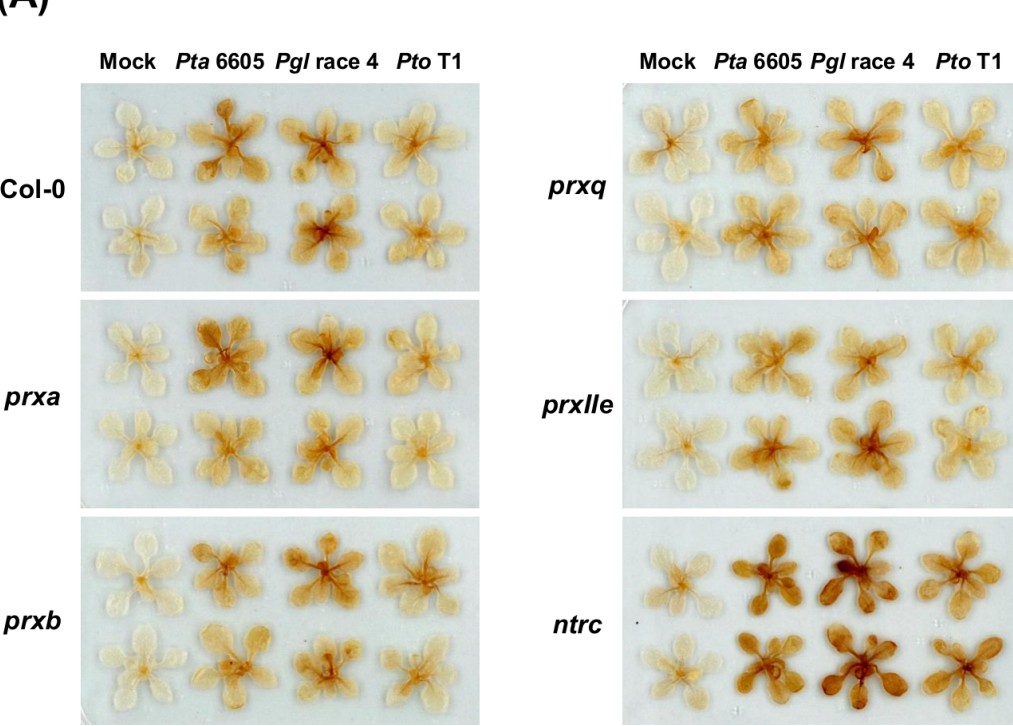

**(B)**

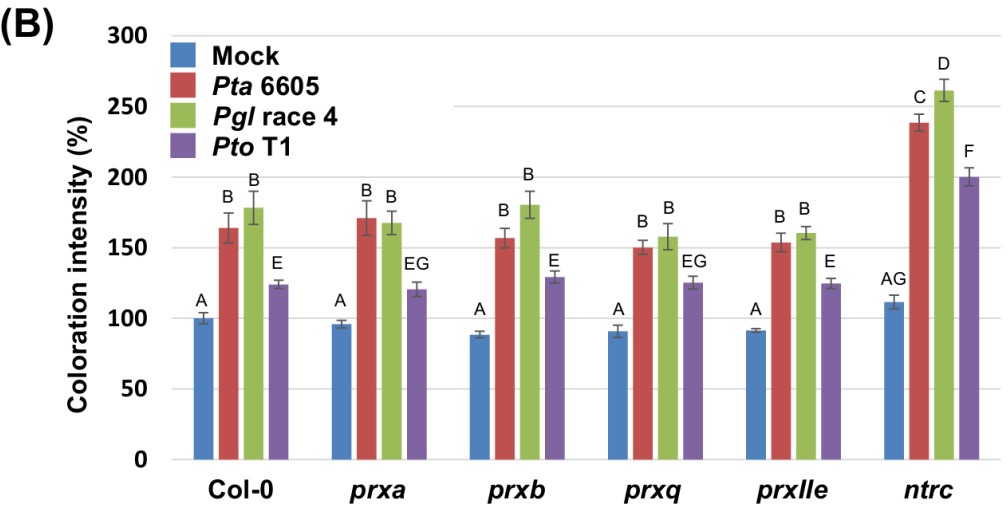

**Figure 2** **Hydrogen peroxide production in the *Arabidopsis* wild-type Col-0, *prx* (*prxa, prxb, prxq,* and *pexIIe*) and *ntrc* mutants in response to *Pseudomonas syringae* pathogens.** (A) Hydrogen peroxide production in *Arabidopsis* chloroplast-localized *prx* and *ntrc* mutants in response to nonhost *P. syringae* pathogens. ROS were visualized by staining hydrogen peroxide using 3,3′-diaminobenzidine at 2 dpi. (B) The intensity of DAB staining shown in panel A is quantified and expressed as a percentage of coloration, where the color intensity of mock-treated wild-type Col-0 leaves was set at 100%. Vertical bars indicate the standard error for 10 leaves. Statistically significant differences are noted as different alphabet characters based on ANOVA ($p < 0.05$).

(*Loehrer et al., 2008*; *Ishiga et al., 2011*; *Lee et al., 2013*), we next assessed the activities of the SA- and JA-mediated signaling pathways by investigating the accumulation of multiple phytohormones such as SA, JA, and JA-Ile, and the expression profiles of the SA- and JA-responsive genes in the wild-type Col-0, the *ntrc* mutant, and the complement line (35S-*NTRC*/*ntrc*).

Two-week-old *Arabidopsis* plants, including the wild-type Col-0, the *ntrc* mutant, and the complement line (35S-*NTRC*/*ntrc*) were grown on MS plates, and were treated with water as a control (mock), or inoculated with *Pta* 6605, and then the samples which were fixed at 24 h after treatment or inoculation were utilized for the quantification of multiple phytohormones by LC-ESI-MS/MS analysis. Interestingly, a higher level of JA and JA-Ile accumulation was observed in the *ntrc* mutant plants in response to the inoculation with *Pta* 6605 compared with the wild-type Col-0 and complement line (Figs. 3B and 3C), whereas no significant differences were observed in the accumulation of SA between the wild-type Col-0 and the *ntrc* mutant (Fig. 3A), indicating that NTRC may negatively regulate the JA biosynthesis pathway.

We next investigated the expression profiles of marker genes for the SA- and JA-signaling pathways, including *ISOCHORISMATE SYNTHASE 1* (*ICS1*), *PATHOGENESIS-RELATED PROTEIN 1* (*PR1*), *LIPOXYGENASE 2* (*LOX2*), *12-OXOPHYTODIENOIC ACID REDUCTASE 3* (*OPR3*), *MYC2*, *GLUTAREDOXIN PROTEIN 480* (*GRX480*), *NAC DOMAIN CONTAINING PROTEIN 19* (*ANAC019*), and *JASMONATE-ZIM-DOMAIN PROTEIN 3* (*JAZ3*) in response to inoculation with *Pta* 6605. Two-week-old *Arabidopsis* plants, including the wild-type Col-0, the *ntrc* mutant, and the complement line (35S-*NTRC*/*ntrc*) were grown on MS plates, and were inoculated with *Pta* 6605, and then total RNA was isolated from samples fixed at 3, 6, 12, and 24 h after inoculation. The expression of SA-mediated signaling pathway genes including *ICS1* and *PR1* was induced by inoculation with *Pta* 6605 among the wild-type Col-0, the *ntrc* mutant, and the complement line (Figs. 4A and 4B). However, lower levels of the *ICS1* and *PR1* transcripts were observed in the *ntrc* mutant compared with the wild-type Col-0 and the complement line (Figs. 4A and 4B). In addition, the expression of JA biosynthesis genes including *LOX2* and *OPR3*, was induced in the *ntrc* mutant in response to inoculation with *Pta* 6605, but not in the wild-type Col-0 and the complement line (Figs. 4C and 4D). Interestingly, the expression of key regulators in the complex cross-talk between the SA- and JA-mediated signaling pathways including *MYC2*, *GRX480*, *ANAC19*, and *JAZ3* was significantly up-regulated in the *ntrc* mutant in response to inoculation with *Pta* 6605. Taken together, these results indicate that the *ntrc* mutant shows elevated JA-mediated signaling pathways in response to nonhost pathogens.

## DISCUSSION

In this study, we conducted the functional analysis of Prx and NTRC, which function as central players of the chloroplast redox detoxification system in the nonhost disease resistance of *Arabidopsis* against bacterial pathogens, and our results demonstrate the relevance of NTRC in nonhost disease resistance against bacterial pathogens. Nonhost

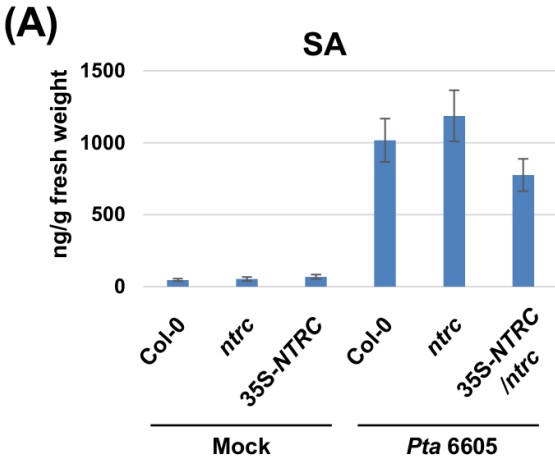

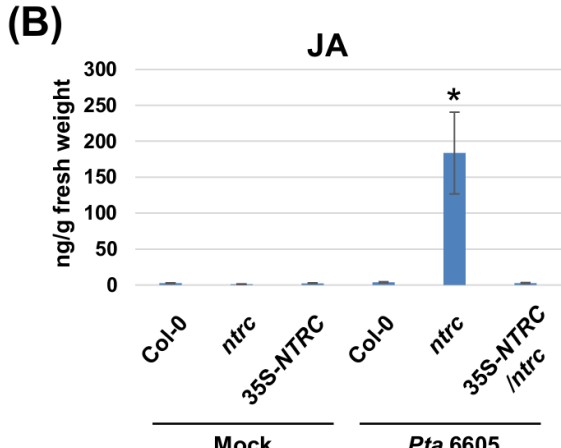

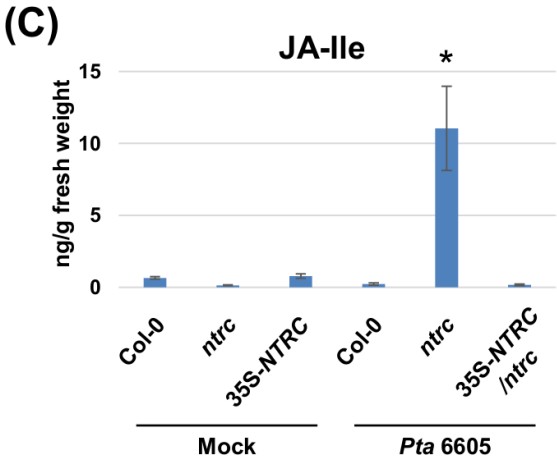

**Figure 3** **Quantification of phytohormones including salicylic acid (SA; A), jasmonic acid (JA; B), and jasmonoyl-L-isoleucine (JA-Ile; C) in the *Arabidopsis* wild-type (Col-0), *ntrc* mutant, and its complemented line (35S-*NTRC*/*ntrc*).** Two-week-old Col-0, *ntrc*, and 35S-*NTRC*/*ntrc* were treated with water as a mock control (Mock) or inoculated with nonhost pathogen *Pseudomonas syringae* pv. *tabaci* (*P. s. tabaci*) at a concentration of $5 \times 10^6$ CFU/ml for 24 h, and phytohormone quantification was performed. Vertical bars indicate the standard errors for four biological replicates. Asterisks indicate a significant difference from the wild-type Col-0 in a *t*-test (* $= p < 0.05$).

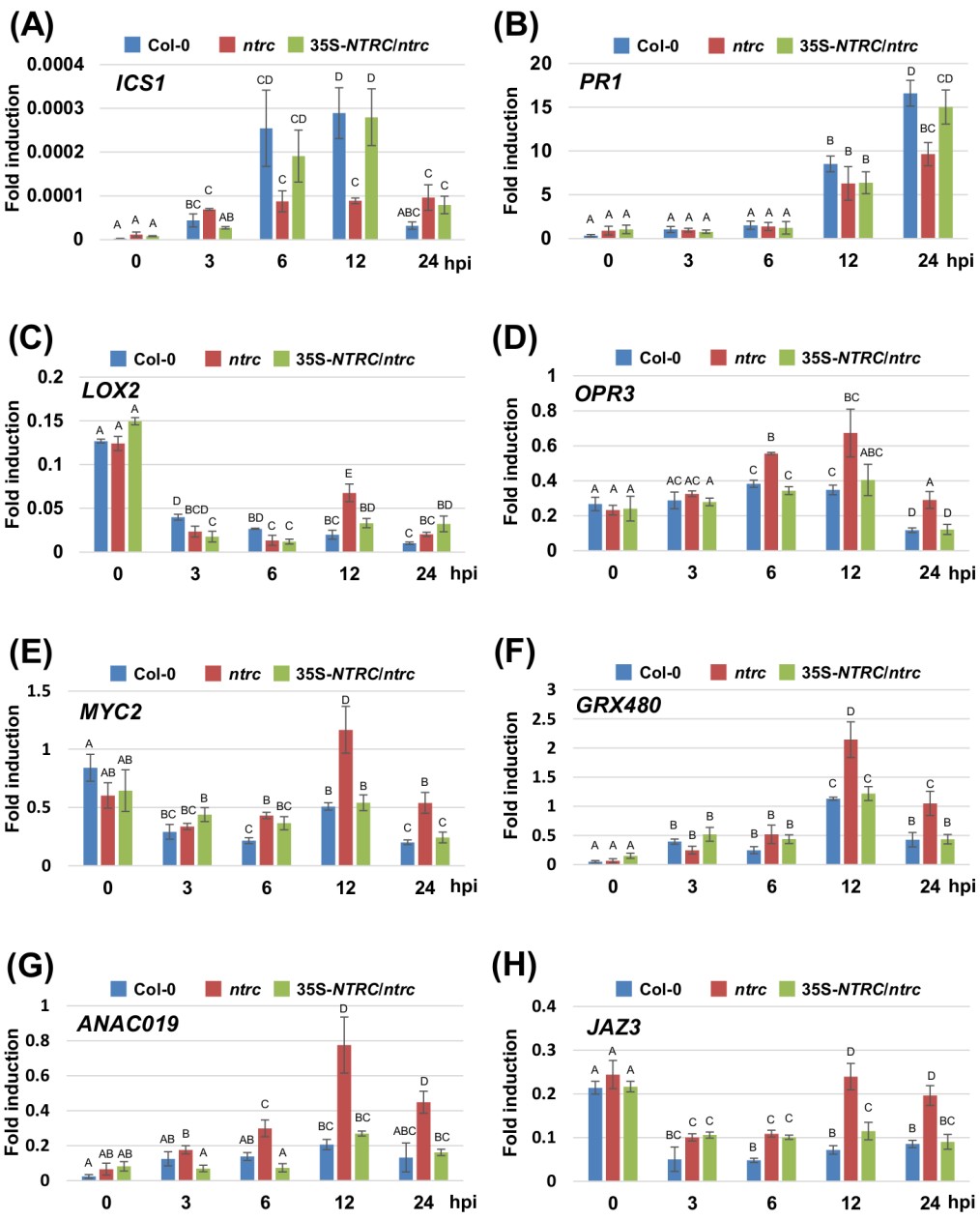

**Figure 4** **Expression profiles of genes encoding for the SA- and JA-mediated signaling pathways in the *Arabidopsis* wild-type (Col-0), *ntrc* mutant, and its complement line (35S-*NTRC/ntrc*).** The expression profiles of the SA-pathway genes including *ISOCHORISMATE SYNTHASE 1* (A) and *PATHOGENESIS-RELATED PROTEIN 1* (B), and JA-pathway genes including *LIPOXYGENASE 2* (C), *12-OXOPHYTODIENOIC ACID REDUCTASE 3* (D), *MYC2* (E), *GLUTAREDOXIN PROTEIN 480* (F), *NAC DOMAIN CONTAINING PROTEIN 19* (G), and *JASMONATE-ZIM-DOMAIN PROTEIN 3* (H) were obtained at 3, 6, 12, and 24 h after inoculation with the nonhost pathogen *Pseudomonas syringae* pv. *tabaci*. Two-week-old *Arabidopsis* Col-0, *ntrc*, and 35S-*NTRC/ntrc* plants were treated with water as a mock control (Mock), or inoculated with nonhost *P. syringae* pv. *tabaci* at a concentration of 5 × 10⁶ CFU/ml. The expression of genes was evaluated by RT-qPCR with gene-specific primer sets (Table S1). The values represent the relative induction compared to the expression of *UBQ1*. Figures are from a representative experiment and repeated at least 3 times with similar results. Vertical bars indicate the standard error for three biological replicates. Statistically significant differences are noted as different alphabet characters based on ANOVA ($p < 0.05$).

disease resistance can be divided into Type I and Type II nonhost resistance based on the presence or absence of visual symptoms. Type I nonhost resistance does not show any visible symptoms and involves PTI, whereas Type II nonhost resistance is associated with visual symptoms because of the HR and involves ETI (*Mysore & Ryu, 2004*; *Niks & Marcel, 2009*; *Fan & Doerner, 2012*; *Senthil-Kumar & Mysore, 2013*). There is overlap between ETI and PTI regarding the signal transduction pathways. In general, ETI is known as a more intense response that is often associated with the HR. The HR involves a wide array of responses including the activation of defense-associated genes, such as pathogenesis-related (PR) genes, phytoalexin production, and the formation of rapid LCD at the site of pathogen invasion to restrict the growth of invading biotrophic and hemibiotrophic pathogens. Both ETI and PTI are known to induce common signal transduction pathways including the oxidative burst and the activation of mitogen-activated protein kinase (MAPK) pathways leading to the expression of defense-associated genes (*Zipfel & Felix, 2005*), indicating that Type I and Type II nonhost resistance involves these signal transduction pathways. In this study, we investigated both Type I and Type II nonhost resistances in the *ntrc* and *prx* mutants against nonhost bacterial pathogens. The results presented in Fig. 1 demonstrated that the *ntrc* mutant showed enhanced LCD not only to Type II nonhost pathogen (*Pta* 6605), but also to Type I nonhost pathogens including *Pto* T1 and *Pgl* race 4 (Figs. 1A and 1B). Interestingly, the LCD observed in the *ntrc* mutant was accompanied by the elevated accumulation of chloroplast-derived ROS (Figs. 2A and 2B), suggesting the correlation between LCD and chloroplast-derived ROS. ROS have been long recognized to orchestrate LCD, and the plasma membrane-bound NADPH oxidase has been considered as the main source of ROS in response to invading pathogens during ETI. However, there is increasing evidence for the role of chloroplast-derived ROS in LCD (*Liu et al., 2007*; *Zurbriggen et al., 2009*). In support of our results, a recent study demonstrated the requirement of chloroplast-derived ROS for the progress of LCD using tobacco plants expressing a chloroplast-targeted cyanobacterial flavodoxin, which functions as a general antioxidant to restrict the formation of ROS in the chloroplasts (*Zurbriggen et al., 2009*). Moreover, they also indicated that chloroplast-derived ROS had no significant effect on the regulation of genes related to defense and photosynthesis. Consistent with these reports, we revealed that the *ntrc* mutant did not show any activation of SA-related genes, such as *ICS1* and *PR1* accompanied by the accumulation of SA in response to nonhost pathogens (Figs. 3A, 4A and 4B). Taken together, these results suggest that chloroplast-derived ROS have an impact on the progress of LCD during the HR, but not on the activation of defense-associated genes. Furthermore, it is likely that this hypothesis could elucidate the phenomena that the ROS marker genes including *sAPX*, *tAPX*, and *GPX1* were not activated in the *ntrc* mutant during nonhost resistance.

In addition, a chloroplast redox homeostasis based on the ferredoxin:thioredoxin system was also shown to regulate LCD in *tomato* (*Lim et al., 2010*). Interestingly, unlike the phenotypes shown in the *ntrc* mutant in this study, *Lim et al. (2010)* showed that silencing of the *tomato ferredoxin:thioredoxin reductase* (*SlFTR-c*) resulted in spontaneous LCD development even without the pathogen inoculation. They also demonstrated that *SlFTR-c*-silenced plants showed enhanced resistance to *Pto* DC3000 by activating the

expression of *PR*-genes (*Lim et al., 2010*). FTR reduces the thioredoxins (TRXs) with electrons derived from ferredoxin, and functions in chloroplast redox homeostasis by regulating chloroplast enzymes. Therefore, it is likely that the silencing of *SlFTR-c* may affect the chloroplast redox homeostasis. Moreover, *Tada et al. (2008)* reported that redox balance change controlled the conformation of NPR1, a master regulator of the SA-mediated signaling pathway leading to the expression of defense-associated genes. They also demonstrated that SA-induced conformation changes in NPR1 were regulated by TRXs (*Tada et al., 2008*). Thus, it is possible that the chloroplast redox change because of the absence of FTR may affect the TRXs-dependent conformation change of NPR1, resulting in the activation of *PR*-genes. In contrast with these results, we showed no obvious activation of the SA-mediated signaling pathways in the *ntrc* mutant. NTRC uses NADPH to detoxify the excessive production of chloroplast-derived ROS coupled with Prx (*Perez-Ruiz et al., 2006*; *Kirchsteiger et al., 2009*; *Bernal-Bayard et al., 2012*; *Bernal-Bayard et al., 2014*; *Puerto-Galán et al., 2015*). Together, these results suggest that although both the FTR and NTRC systems play an important role in the chloroplast redox regulation, the precise functions for these systems may diverge based on the target molecules.

We showed that the loss-of-function of *NTRC* resulted in excessive ROS accumulation when compared to the wild-type in response to nonhost bacterial pathogens (Fig. 2), causing enhanced LCD. Furthermore, the *ntrc* mutant showed elevated growth of nonhost bacterial pathogens (Fig. 1). Therefore, one could argue how enhanced LCD can contribute to bacterial growth in the *ntrc* mutant. One possibility that should be considered is that *P. syringae* pathovars could get nutrients from the tissues showing LCD during the necrotrophic growth phase. *P. syringae* is a hemibiotrophic bacterial pathogen that shows no symptoms during the biotrophic growth phase including epiphytic colonization on the leaf surface and early apoplastic colonization. However, once the bacterial population density grows to a particular quorum-sensing (QS) threshold, QS molecules trigger the expression of a large number of *P. syringae* genes, which regulate the formation of disease-associated cell death, resulting in a switch to the necrotrophic growth phase (*Newton et al., 2010*). Although it has been considered that LCD is more effective in plant defense systems against invading obligate biotrophic pathogens, LCD may enable necrotrophic pathogens to multiply by providing nutrients from dead tissues (*Mur et al., 2008*; *Coll, Epple & Dangl, 2011*). It has been reported that some necrotrophic pathogens could promote the ROS production. In addition, several *P. syringae* pathovars are known to produce nonhost-specific phytotoxins to promote lesions associated with symptoms. For example, *Pgl* and *Pta* produce the chlorosis-inducing phytotoxins coronatine (COR) and tabtoxin, respectively, for their pathogenicity (*Bender, Alarcón-Chaidez & Gross, 1999*). Interestingly, in our previous study (*Ishiga et al., 2009*) and an ultrastructural study by *Palmer & Bender (1995)* revealed that COR targets the chloroplast for pathogenicity (*Palmer & Bender, 1995*; *Ishiga et al., 2009*). Palmer and Bender also demonstrated that the apoplastic extracts from tomato leaves treated with COR could enhance the multiplication of *Pto*, indicating that the leakage of nutrients from COR-treated tissues could be sufficient for *Pto* growth (*Palmer & Bender, 1995*). Furthermore, it has been well documented that bacterial pathogens can multiply during epiphytic and apoplastic colonization, even in nonhost plants

(*Ishiga et al., 2011*; *Rojas et al., 2012*; *Senthil-Kumar & Mysore, 2012*). Consistent with previous reports, the endophytic bacterial multiplication was detected in the *Arabidopsis* Col-0 wild-type inoculated with nonhost pathogens (Figs. 1A–1C), suggesting that these bacterial pathogens may have potential for multiplication to a certain level in the apoplastic space of nonhost plants. Therefore, it is tempting to speculate that the nutrients derived from tissues showing enhanced LCD enable *P. syringae* pathovars to multiply during the necrotrophic growth phase.

Our expression profiles and phytohormone analyses revealed the activation of genes related to the JA biosynthesis pathway, including *LOX2* and *OPR3* (Figs. 4C and 4D), and the concomitant accumulation of JA and JA-Ile in the *ntrc* mutant in response to a nonhost bacterial pathogen (Figs. 3B and 3C). JA-mediated signaling pathways have been well characterized because of their importance in plant defense systems in biotic and abiotic stress response. JA is known to activate the plant defense system in response to wounding, herbivory, attack by necrotrophic pathogens, and environmental stresses including low temperature, salinity, and drought (*Campos, Kang & Howe, 2014*; *Savatin et al., 2014*; *Riemann et al., 2015*; *Sharma & Laxmi, 2015*). Consistent with our results, *Demmig-Adams et al. (2013)* showed that *Arabidopsis* knockout mutants of key components of the chloroplast photoprotection system produced high levels of JA and its precursor 12-oxo phytodienoic acid (OPDA). Moreover, it has been reported that the expression of genes related to the JA-mediated signaling pathway were induced in response to high light stress in *Arabidopsis* (*Tikkanen et al., 2014*). It is well-known that abiotic stresses including high light induce the generation of ROS in the chloroplast (*Suzuki et al., 2012*). Therefore, the chloroplast is considered to contain a large amount of polyunsaturated fatty acids, the precursors for JA and OPDA, which show sensitivity to oxidation by ROS (*Schaller & Stintzi, 2009*). Together, our results suggest the correlation between chloroplast-derived ROS and the JA-signaling pathway in the *ntrc* mutant in response to a nonhost bacterial pathogen.

We also demonstrated the induction of genes related to key regulators for the JA-mediated signaling pathway, including *MYC2*, *GRX480*, *ANAC019*, and *JAZ3* in the *ntrc* mutant in response to a nonhost bacterial pathogen (Figs. 4E–4H), indicating that the loss-of-function of *NTRC* resulted in the activation of the JA-mediated signaling pathway during nonhost disease resistance. It has been well studied that cross-talk between SA and JA plays an important role in fine tuning plant defenses to enhance plant fitness, and that MYC2 functions as a key regulator of cross-talk between SA and JA (*Laurie-Berry et al., 2006*; *Pieterse et al., 2009*; *Thaler, Humphrey & Whiteman, 2012*). *Zheng et al. (2012)* reported that MYC2 bound to the promoter of the NAC transcription factor *ANAC019* and activated its expression. Furthermore, ANAC019 was shown to suppress the expression of *ICS*. Consistent with these reports, the expression profiles of SA-mediated signaling revealed lower levels of the *ICS1* and *PR1* transcripts in the *ntrc* mutant compared to the wild-type Col-0 during nonhost resistance (Figs. 4A and 4B). In addition, our previous studies showed that the SA-mediated signaling pathway played an important role in nonhost disease resistance against bacterial pathogens (*Ishiga et al., 2011*). However, our phytohormone analyses showed that there were no significant differences in SA accumulation between the wild-type Col-0 and the *ntrc* mutant (Fig. 3A). Therefore, it is

possible that chloroplast-derived ROS may function as a signaling compound to regulate the JA-pathway, but not the SA-pathway.

## CONCLUSIONS

This study characterized the function of NTRC with respect to the nonhost disease resistance of *Arabidopsis* against bacterial pathogens, and provides additional insights into the mechanism of plant innate immunity, especially the correlation between chloroplast-derived ROS and LCD. Further investigation of *ntrc* mutant together with SA or JA related mutants will be required to fully understand the complex mechanisms of nonhost disease resistance with reference to NTRC.

## ACKNOWLEDGEMENTS

We thank Dr. Christina Baker for editing the manuscript.

### Funding

This work was supported, in part, by the Program to Disseminate Tenure Tracking System, MEXT, Japan, and the Samuel Roberts Noble Foundation. This work was also supported by the Japan Advanced Plant Science Network. The funders had no role in study design, data collection and analysis, decision to publish, or preparation of the manuscript.

### Grant Disclosures

The following grant information was disclosed by the authors:
Samuel Roberts Noble Foundation.
Japan Advanced Plant Science Network.

### Competing Interests

Kirankumar S Mysore is an Academic Editor for PeerJ.

### Author Contributions

- Yasuhiro Ishiga and Takako Ishiga conceived and designed the experiments, performed the experiments, analyzed the data, contributed reagents/materials/analysis tools, wrote the paper, prepared figures and/or tables, reviewed drafts of the paper.
- Yoko Ikeda performed the experiments, analyzed the data, contributed reagents/materials/analysis tools, wrote the paper, prepared figures and/or tables, reviewed drafts of the paper.
- Takakazu Matsuura performed the experiments, analyzed the data, contributed reagents/materials/analysis tools, prepared figures and/or tables, reviewed drafts of the paper.
- Kirankumar S. Mysore conceived and designed the experiments, contributed reagents/materials/analysis tools, wrote the paper, prepared figures and/or tables, reviewed drafts of the paper.
## Data Availability

Figshare: http://figshare.com/s/d606fc109c8d11e5b66906ec4b8d1f61.

## Supplemental Information

Supplemental information for this article can be found online at http://dx.doi.org/10.7717/peerj.1938#supplemental-information.

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
