# Peer review of "NADPH-dependent thioredoxin reductase C plays a role in nonhost disease resistance against Pseudomonas syringae pathogens by regulating chloroplast-generated reactive oxygen species"

_PeerJ, doi:10.7717/peerj.1938_

## Round 0.1 · original submission · Major Revisions

· Academic Editor

Major Revisions

After carefully reading your manuscript and in agreement with both reviewers I find your research question is well defined and merits investigation. The methods are adequate and your provided data are statistically sound. However, I have different major concerns, also addressed by both reviewers, that point to a rigorous revision of your manuscript.

As indicated by reviewer 1 the fundamental concern is the fact that your major conclusions were already reported in a previous report. Since replication experiments are encouraged in PeerJ, it is absolutely necessary that you clearly provide the rationale for the replication, and how it adds value to the literature. Furthermore, you should clearly address the major objection that only GPX7 is induced in the ntrc-mutant and no further ROS marker gene, what makes your conclusions speculative.

In addition and as suggested by reviewer 2, a more thorough description of the HR should be included in the introduction. Moreover, a complete reformulation of the discussion is strongly recommended, including the points raised by this reviewer. It should be considered that localized cell death (LCD) is a response of the plant to the attack of an invading pathogen and ROS from various cellular sources have been associated with promotion of LCD. This manuscript shows that elimination of NTRC and subsequent ROS build-up leads to increased pathogen propagation, which contradicts many previous observations. This observation is therefore unexpected and should be discussed in more depth.

Reviewer 1 ·

Basic reporting

The manuscript by Ishiga et al. describes the role of NTRC and plastidial Prxs (A, B, Q and IIE) in chloroplast ROS production and the effect on disease resistance against Pseudomonas syringae. While Prx-deficient mutants of Arabidopsis behave like wild type plants in response to P. syringae pathogens, the ntrc mutant shows increased susceptibility (ion leakage, growth a bacteria), as well as higher production of hydrogen peroxide. Based on the analysis of the expression patterns of genes responsive to oxidative stress, SA and JA authors conclude that the ntrc mutant has impaired cross-talk between JA and SA signaling pathways and, thus, in plant innate immunity.

Experimental design

Adequate

Validity of the findings

The manuscript is well written and results are clearly described. However, I find a major limitation of this study which is the fact that the major conclusions were already reported by at least some of the authors in a previous report (Ishiga et al (2012) MPMI). In this paper the prx and ntrc mutants were used to show the involvement of NTRC (not of Prxs) in response tp P.syringae pv tomato. The present study goes on analyzing SA and JA pathways in the ntrc mutant and other pathovars, but I think this is not a substantial advance. In addition, conclusions are somehow speculative and suggest that the present study is still preliminary.
Additional points
1. On lines 216-217 based on the lack of symptom development in prx mutants, authors suggest redundant functions for these Prxs. In my opinion the results do not allow to make such suggestions. It is well known that typical 2-Cys Prxs (A and B) are quite different from atypical 2-Cys Prx Q and IIE. For example NTRC is a good reductant of 2-Cys Prxs but not of the atypical ones.
2. Among the ROS marker genes analyzed (sAPX, tAPX, GPX1 and GPX7) only GPX7 was induced in the ntrc mutant (lines 261-265). This result is poorly supportive to suggest that the cell death phenotype of the ntrc mutant is due to ROS accumulation. Similarly, the proposal of the important role of NTRC in cell death and ROS homeostasis requires more experimental support.
3. Finally, the final part of the Discussion indicating the role that NTRC might play in innate immunity by regulating stromule formation is again highly speculative and requires more experimental support.

Comments for the author

See above comments

Reviewer 2 ·

Basic reporting

The article “NADPH-dependent thioredoxin reductase C plays a role in nonhost disease resistance against Pseudomonas syringae pathogens by regulating chloroplast-generated reactive oxygen species (#7983)” by Ishiga et al. describes the response of KO Arabidopsis mutants in individual genes of the chloroplast peroxiredoxin (Prx) and NADPH-dependent thioredoxin reductase C (ntrc) detoxification system to infection with three different non-host pathogens of the genus Pseudomonas.
The article is written in correct English and, with some exceptions, can be followed easily. The intro and background place the research in the context of the theoretical framework for non-host interactions advanced by one of the authors a few years ago. The differences between PTI and ETI are only briefly addressed and may lead to confusion, but this issue is not relevant to the subject of the paper. Instead, I believe a more thorough description of the HR should be included in the intro, because I find that it is important for discussion of the results (see below).
Figures are adequate and are well labeled and quoted in the text.

Experimental design

The problem is well defined and the nature of the question deserves to be investigated. Methods are adequate for the purpose of the research. Data are convincing and statistically sound.

Validity of the findings

As indicated in the previous section, the problem is well defined and the nature of the question deserves to be investigated. Methods are adequate for the purpose of the research. Data are convincing and statistically sound.
I have, however, some concerns with the discussion section. First, it is generally accepted, and extensively documented throughout the scientific literature, that localized cell death (LCD) is a response of the plant to the attack of an invading pathogen that results in suicidal cell death around the site of infection. This is expected to deter the spread of biotrophic and hemibiotrophic microorganisms which require living tissue to propagate. Also, ROS from various cellular sources have been associated, in different ways, with promotion of LCD. Results obtained with FTR (and others quoted by the authors) agree with these tenets. In contrast, the reports of this paper indicate that elimination of a chloroplast antioxidant (NTRc) leads to ROS build-up (which seems reasonable), but also to increased pathogen propagation. Then, in the case of the ntrc KO, LCD favors spread of the microorganism, which contradicts many previous observations. This observation is therefore unexpected and should be discussed in more depth to provide hints on the reasons of the discrepancy.
When attempting to do so, the authors indicate that JA levels are up-regulated in the mutants. JA is known to be involved in resistance to necrotrophs but it also accumulates in other stress situations, both biotic and abiotic. However, the discussion of the possible contribution of JA to the effects observed is very short (lines 331 to 353). Subsequently, the authors engage in a long discussion on the involvement of SA on biotic stresses (also extensively documented in the literature), that extends from line 354 on. But SA levels do not change in the mutant relative to the WT! Then, the discussion becomes weak and cursory. I strongly recommend that the authors reformulate their discussion, considering that the journal welcomes speculation. A stronger discussion should include (at least): i) a better description of LCD associated with the HR, ii) the possible role of chloroplast ROS in LCD, iii) how can LCD help pathogen propagation in the ntrc mutants?, iv) could JA antagonize SA signal transduction in this system, as has been proposed earlier (and questioned)?, v) could NTRc play roles different from those of FTR besides their obvious differences under light and dark conditions?

Comments for the author

Minor comments
1. The authors should consider the possibility of using abbreviations to name the pathovars (e. g., Psto, Psta, Psgl, or something similar). The full names included in several places makes reading unrewarding.
2. Line 119, the ecotype is Columbia, not Colombia.
3. Line 219, with respect to WT Arabidopsis, Ps glycinea and tomato T1 conform to the type I non host resistance defined above, in the same page, by the authors. Not to type II as indicated. Actually, the ntrc mutation transforms a type I phenotype into a type II.
4. Line 223, I would not say that Ps tabaci induces a MUCH stronger CD. While statistically significant, differences shown in Figure 1B are in the order of 10% or less.

---

## Round 0.2 · accepted · Accept

· Academic Editor

Accept

After carefully reading your revised manuscript and in agreement with the reviewer I came to the conclusion that you addressed all points raised by the reviewer. You adequately responded all concerns, improved the introduction, and in particular, rigorously revised the discussion. These amendments improved the manuscripts´ readability and content.

Reviewer 2 ·

Basic reporting

My assessment of the quality and relevance of the research has been provided with the first evaluation. At this stage, I believe the authors have properly responded my concerns and added valuable information that enhances the impact and readability of the text.

Experimental design

See above.

Validity of the findings

See above.

Comments for the author

No comments.